# Low-Profile High-Efficiency Transmitarray Antenna for Beamforming Applications

**Jae-Gon Lee [1,*]** and **Jeong-Hae Lee [2,*]**

1    Department of Electronic and Software Engineering, Kyungnam University, Changwon 51767, Republic of Korea

2    Department of Electronic and Electrical Engineering, Hongik University, Seoul 04066, Republic of Korea

\*    Correspondence: jaegonlee@kyungnam.ac.kr (J.-G.L.); jeonglee@hongik.ac.kr (J.-H.L.)

**Abstract:** A low-profile high-efficiency transmitarray antenna (TA) for beamforming applications is proposed and investigated in this paper. The partial H-plane waveguide slot array antenna is employed as the compact low-profile feeding structure of the beamforming TA. The designed TA can achieve a high taper efficiency due to the multi-array sources and the compactness of the partial H-plane waveguide. Moreover, the proposed TA can inherently have a high spillover efficiency because the frequency selective surface (FSS) for beamforming is located just above the radiating slot. The FSS with a transmission phase variation of $2\pi$ is designed by a square patch array and used to manipulate the wave-front of the transmitted electromagnetic wave instead of a complicated feed network and phase shifters. To verify its beamforming characteristic, three types of FSSs to operate a forming angle of $-40°$, $-20°$, $0°$, $+20°$, and $+40°$ are designed at 12 GHz. The distance between the FSS and the slot array antenna is $0.1\lambda_0$, and the aperture efficiency is measured to be about 69%. The measured results, such as the reflection coefficient and the far-field radiation pattern, are in good agreement with the simulated results. From the measured results, the proposed TA is confirmed to have good beamforming characteristics and high aperture efficiency.

**Keywords:** beamforming; low profile; high efficiency; transmitarray antenna (TA)

## 1. Introduction

High-directivity beamforming and beam-steering antennas are very attractive and useful in wireless technology, especially in 5G wireless communication, wireless power transfer (WPT), and satellite communication. The most common method for beam control is to employ a phased array antenna. The phased array antenna is composed of many radiating elements and many phase shifters. Beams are formed by shifting the phase of the signal radiating from each radiating element to provide constructive interference to steer the beams toward the desired direction. Many types of phased array antennas can achieve good performances in various applications [1–7]. However, the phased array antennas have a complicated feed network with a loss that cannot be ignored in most applications requiring high gain as the number of radiating elements increases. Moreover, the employment of many phase shifters gives rise to high costs and additional losses.

To overcome these drawbacks of the conventional phased array antenna, transmitarray antennas (TAs) have recently been researched as an alternative to the phased array antenna [8–16]. TAs composed of a single source antenna and a planar metasurface (MS) can be designed by controlling the phase and amplitude of the MS. To obtain a good beamforming characteristic, the MS should have good transmittance and a full transmission phase variation of $2\pi$. To achieve beam steering at the desired direction, the shape of the transmission phase should be controlled using an active MS loaded by varactor, diode, MEMS, piezoelectric, ferroelectric, liquid crystal (LC), and so on, and the shape of the transmission phase should be controlled by changing the phase front of the feed source.

To design a highly efficiency TA, there are two structurally important parameters, such as the diameter (*D*) of the MS and the distance (*F*) from the feed antenna to the MS. The spillover and taper efficiencies, which are determined by the amount of power reaching the MS from the feed source and their uniformity, respectively, depend on the *F*/*D* ratio. As the diameter of an MS increases, the spillover efficiency increases. Conversely, as the distance of *F* increases, the taper (spillover) efficiency increases (decreases), respectively. Therefore, *F*/*D* should be optimized to achieve high aperture efficiency. Most recently studied TAs have an *F*/*D* ratio of 0.2 to 1, and *F* is greater than $1\lambda_0$. Therefore, many TAs are inherently bulky and cannot be designed with a low profile. Another type of TA has a multi-array source antenna and MS. Even if the MS is placed close to the source antenna, this type of TA can inherently have high spillover efficiency and high taper efficiency because of the uniform power distribution to reach the MS. As a result, a low-profile beam-steering TA can be designed using a transmissive phase gradient MS. In [17], a conventional waveguide slot array antenna is utilized to achieve the low-profile beam-steering TA for satellite communication. To obtain a high directivity, large areas to implement many radiating slots are required.

In this paper, a low-profile high-efficiency TA for beamforming applications is proposed and investigated. The partial H-plane waveguide slot array antenna is utilized as the compact low-profile feeding structure of the beamforming TA. The partial H-plane waveguide is a transversely folded rectangular waveguide that has a partially inserted metal vane in the H-plane [18]. As the partial H-plane waveguide is more compact than the conventional waveguide, the number of radiating slots of the partial H-plane waveguide slot array antenna is larger than that of the conventional waveguide slot array antenna in the same design area. Therefore, the partial H-plane waveguide slot array antenna can have a higher taper efficiency compared to the conventional waveguide slot array antenna. To design an 8 × 10 array antenna, a series type of 8-way power divider for the partial H-plane waveguide slot array antenna is proposed. An FSS, that is, any thin and repetitive surface designed to reflect, transmit, or absorb electromagnetic waves, is employed to control the phase and amplitude of the source antenna [19,20]. In particular, the FSS is designed by a square patch array based on a metallo-dielectric FSS [21] and is formed by four dielectric layers with equally spaced capacitive printed patches. Each unit cell of the FSS is fed by each slot of a partial H-plane waveguide slot array antenna and is symmetrically located above the slot. The phase set required for beamforming is controlled by the unit cell dimension of the FSS located above each slot. To confirm its beamforming performance, three types of FSSs to operate a forming angle of −40°, −20°, 0°, +20°, and +40° are designed and integrated into the slot array antenna at 12 GHz. The organization of this paper is as follows. In Section 2, the design procedure of the two-dimensional (2D) partial H-plane waveguide slot array antenna is described. In Section 3, the unit cell of the FSS is theoretically analyzed, and its transmittance characteristic is confirmed by a full-wave simulation. Moreover, the beamforming performance of the partial H-plane waveguide slot array antenna integrated with the FSS is verified by simulated and measured results. Finally, the conclusion is presented in Section 4.

## 2. Design of 2D Partial H-Plane Waveguide Slot Array Antenna

The compact partial H-plane waveguide in Figure 1 was proposed in [18]. The propagation constant ($\beta_y$) of a longitudinal direction and cut-off frequency ($f_c$) can be calculated by the following [18]:

$$\beta_y = \sqrt{\beta_0^2 - \beta_{z1}^2 - \left(\frac{2m\pi}{w}\right)^2} \quad (m = 0,\ 1,\ 2,\ \cdots) \tag{1}$$

$$f_c = \frac{c}{2\pi}\sqrt{\beta_{z1}^2 + \left(\frac{2m\pi}{w}\right)^2} \quad (m = 0,\ 1,\ 2,\ \cdots) \tag{2}$$

where $c$ and $w$ are the velocity of light and the width of the partial H-plane waveguide, respectively. $\beta_0$ and $\beta_{z1}$ are the wave number in the air and the wave number of the z-direction in the region 1, respectively.

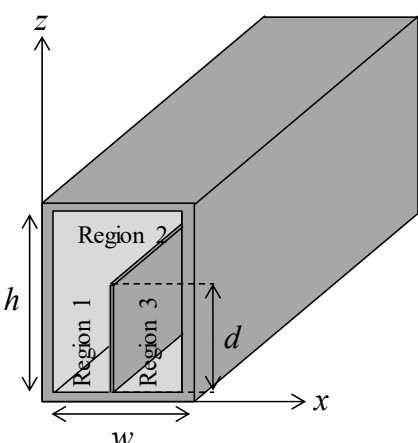

**Figure 1.** Structure of the partial H-plane waveguide.

Figure 2a shows the structure of the $1 \times 10$ partial H-plane waveguide slot array antenna. The slot length is nearly $\lambda_0/2$, and its width is assumed to be small. To be excited in phase for all the slots, the array with slots spaced $\lambda_g/2$ apart and with alternating slots on the opposite side of the center line is employed [22,23]. The longitudinal shunt slot does not radiate when the slot is located at the centered line because the transverse current and the longitudinal magnetic field are zero in the dominant mode. The longitudinal magnetic field symmetrically increases with offset distance from the centered line of the top plane of a partial H-plane waveguide. The slot offset is determined to be at a maximum value of 4.25 mm considering a fabrication process so that maximum excitation level at each slot can be achieved. The extended ground is designed to place the unit cell of an FSS on the slot symmetrically. The unit cell of an FSS is described in detail in Section 3. The dimensions of the designed $1 \times 10$ partial H-plane waveguide slot array antenna operating at 12 GHz are listed in Table 1. In summary, the slot length ($l$) and the slot periodicity ($p$) are nearly $\lambda_0/2$ and $\lambda_g/2$, respectively, and are optimized for in-phase excitation at all slots. The slot width is 0.35 mm in this paper, so that can be assumed to be small. Also, when the slot offset ($x$) increases, the aperture efficiency is improved. Figure 2b shows the structure of the $8 \times 10$ partial H-plane waveguide slot array antenna with power divider operating at 12 GHz. To maintain the electric and magnetic field distribution in the divided partial H-plane waveguide, the vane of the input port of the power divider is split and connected to the output ports. The $8 \times 10$ partial H-plane waveguide slot array antenna is designed using a seven two-way power divider, as shown in Figure 2b. To excite the dominant mode of the designed array antenna, the coaxial transition as a feeding structure is used considering a fabrication. A coaxial probe from the narrow sidewall is inserted into the rectangular intaglio in the metal vane, and it is located at about a quarter wavelength long distance from the end metal wall. After optimizing the feeding structure, we find the dimension of a rectangular intaglio in the metal vane. The width and height of the rectangular intaglio are 3.4 mm and 2.26 mm, respectively. We confirm that all transmission is achieved, and the dominant mode is excited at partial H-plane waveguide at 12 GHz. The full-wave simulated reflection coefficient and realized gain of the designed array antenna are presented in Figure 3a and b, respectively. The simulated reflection coefficient is $-23.67$ dB at the operation frequency, and the $-10$ dB fractional bandwidth is 0.93%. The simulated maximum gain of the designed slot array antenna is 25.3 dBi at 12 GHz. As the width of the partial H-plane waveguide is inherently smaller than that of the conventional waveguide, the partial H-plane waveguide slot array antenna can have more radiating slots than the conventional waveguide slot array antenna in the same design area. Therefore,

the taper efficiency ($\eta_t$) of the partial H-plane waveguide slot array antenna can be higher than that of the conventional waveguide slot array antenna. To confirm this property, we calculate and compare their taper efficiency in the plane where the FSS is located using Equation (3), as follows:

$$\eta_t = \frac{1}{S} \cdot \frac{\left| \int E(x,y) dS \right|^2}{\int |E(x,y)|^2 dS} \tag{3}$$

where $S$ is the area of the FSS [24]. The tangential electric field of the area where the FSS is placed is calculated by the commercial ANSYS Electronics desktop software. The design area of the $8 \times 10$ partial H-plane waveguide slot array antenna is nearly the same as that of the $6 \times 10$ waveguide slot array antenna. The simulated taper efficiencies of the designed $8 \times 10$ partial H-plane waveguide slot array antenna and the $6 \times 10$ conventional waveguide slot array antenna are 77% and 71%, respectively.

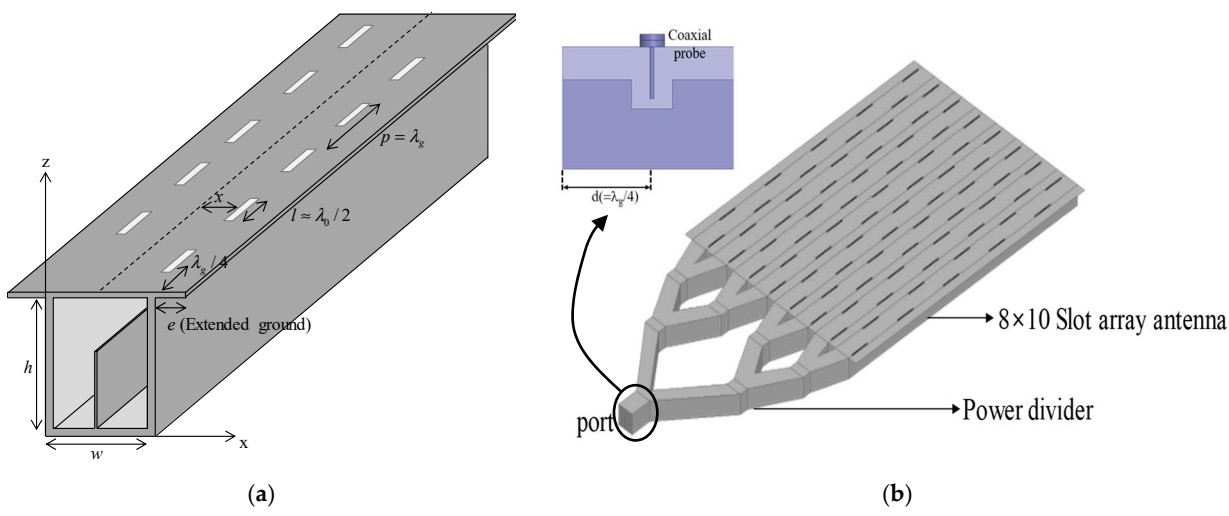

(**a**)　　　　　　　　　　　　　　　　　　　　　　　　　　　(**b**)

**Figure 2.** Structure of the partial H-plane waveguide slot array antenna. (**a**) $1 \times 10$ array and (**b**) $8 \times 10$ array with power divider.

**Table 1.** Dimensions of the designed $1 \times 10$ partial H-plane waveguide slot array antenna (unit: mm).

| Cross Section ($w \times h$) | Slot Length ($l$) | Slot Periodicity ($p$) | Offset ($x$) | Length of Extended Ground ($e$) |
|---|---|---|---|---|
| $9.43 \times 9.43$ | 11.2 | 16.145 | 4.25 | 2.5 |

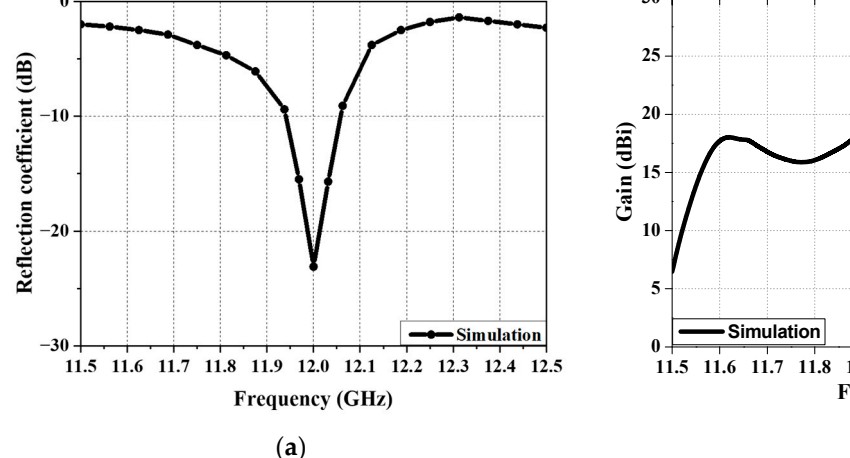

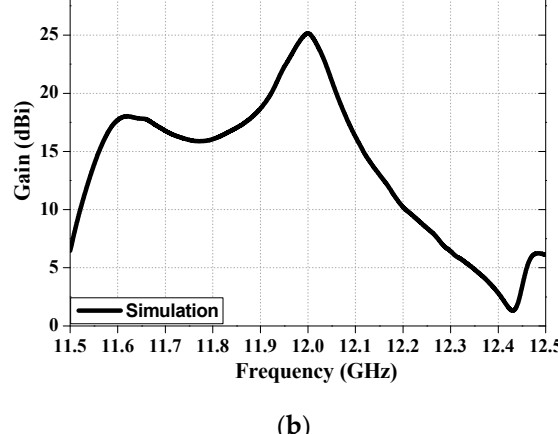

(**a**)　　　　　　　　　　　　　　　　　　　　　　　　　　　(**b**)

**Figure 3.** Full-wave simulated results. (**a**) Reflection coefficient and (**b**) realized gain versus frequency.

## 3. Beamforming Partial H-Plane Waveguide Slot Array Antenna

To design the FSS for the proposed beamforming slot array antenna, a square patch array is presented in this paper. The unit cell is based on a metallo-dielectric FSS [21] and formed by three dielectric layers with equally spaced capacitive square-printed patches, as shown in Figure 4. The patch array FSS has the characteristic of a low-pass filter and can have a transmission phase variation of $2\pi$, which can be controlled by the dimension of a patch in the passband. If the TA has multi-array sources, both spillover and taper efficiencies are inversely proportional to $F/D$ ratio. When distance ($F$) between the FSS and the slot array antenna becomes narrow, the aperture efficiency is improved. However, the fabricated FSS has to be integrated with the 2D partial H-plane waveguide array antenna, and a minimum space for the supporter is required. As a result, the distance ($F$) is chosen as 2.5 mm ($0.1\lambda_0$ at 12 GHz). Figure 5 shows the simulation setup to obtain the transmittance response in the commercial ANSYS Electronics desktop software. The operation frequency of the FSS is 12 GHz, and the utilized substrate for the FSS is TLY-5 ($\varepsilon_r$ = 2.2 and tan δ = 0.0009). The thickness of one substrate ($hm$) and the dimension ($wm$) of the unit cell are 0.8 mm and 14.43 mm, respectively. The transmittance can be obtained by $1-\Gamma^2$ ($\Gamma$ = reflection coefficient at port), and the transmission phase can be calculated by the difference in the transverse electric field through the FSS. The unit cell is symmetrically located above the slot, and the dimension of the unit cell is designed to be equal to the slot periodicity to obtain the same transmission characteristics for the slot source. Table 2 shows the transmittance response of the optimized FSS against patch dimension. When the dimension of a patch is larger than 7 mm, the transmittance is less than 0.9 and poorer than those of others. Thus, as $pw_2$ is larger than 7 mm, we simulated and computed transmittances and transmission phases by changing $pw_1$ with fixed $pw_2$, where $pw_1$ is the dimension of the first and fourth patches, and $pw_2$ is the dimension of the second and third patches. As a result, the unit cell has a good transmittance (>0.9) and a full transmission phase variation of about 530° by changing the dimensions of the patches, as shown in Table 2.

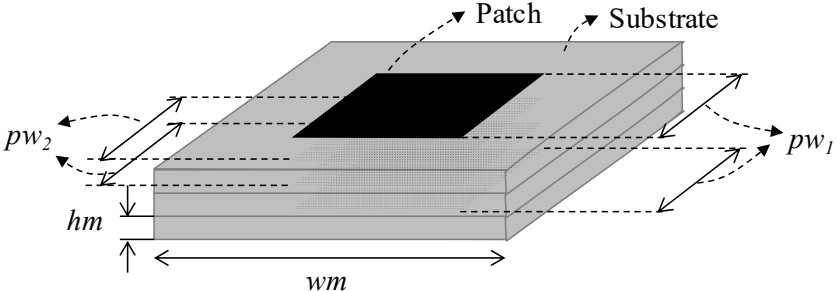

**Figure 4.** Unit cell of the designed frequency selective surface (FSS).

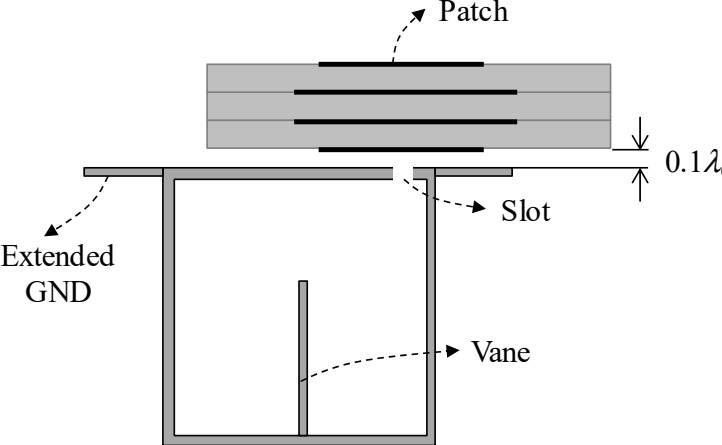

**Figure 5.** Simulation setup for the transmittance response of the unit cell of the FSS.

**Table 2.** Transmittance responses of the designed FSS against patch dimensions (unit: mm).

| $pw_1$ | 0.5 | 1 | 1.5 | 2 | 2.5 | 3 | 3.5 | 4 |
|---|---|---|---|---|---|---|---|---|
| $pw_2$ | 0.5 | 1 | 1.5 | 2 | 2.5 | 3 | 3.5 | 4 |
| Transmittance (Linear) | 0.91 | 0.9 | 0.91 | 0.93 | 0.95 | 0.97 | 0.99 | 0.99 |
| Transmission phase (Degree) | −73 | −76.9 | −77.5 | −81.8 | −88 | −96.6 | −106.3 | −120.9 |
| $pw_1$ | 4.5 | 5 | 5.5 | 6 | 6.5 | 6.6 | 6.9 | 7.3 |
| $pw_2$ | 4.5 | 5 | 5.5 | 6 | 6.5 | 7 | 7.5 | 7.7 |
| Transmittance (Linear) | 0.99 | 0.97 | 0.95 | 0.95 | 0.98 | 0.99 | 0.97 | 0.9 |
| Transmission phase (Degree) | −134.1 | −150.6 | −162.1 | −170.8 | −181.6 | −252.6 | −404.9 | −604.5 |

Figure 6 shows the structure of the proposed beamforming partial H-plane waveguide slot array antenna. The distance between the FSS and the 2D partial H-plane waveguide array antenna is $0.1\lambda_0$. The overall dimension of the proposed beamforming array antenna is 120.44 mm (width) × 261.45 mm (length) × 9.43 mm (height). To achieve the 1D beamforming in the $x$–$z$ plane, the array factor (AF) of a uniform amplitude and spacing can be considered because the magnitudes of the incident field at the unit cell of the FSS are nearly identical. When the array distance is $d_x$ in the $x$-axis, the AF of the array is expressed as

$$\text{AF} = \sum_{n=1}^{N} exp\{j(n-1)(kd_x cos\theta + \beta_x)\} \tag{4}$$

where $N$ and $\beta_x$ are the number of arrays and the phase difference between the adjacent units in the $x$-direction, respectively [24]. The phase difference for radiating toward a specific angle can be calculated from Equation (4) and controlled by the designed FSS. The dimensions of the cells of the FSS for eight rows against the beamforming angle of $0°$, $+20°$, and $+40°$ are listed in Table 3. Similarly, the dimensions of the cells of the FSS for $−20°$ and $−40°$ are symmetrical to the $y$-axis. Figure 7 shows the photographs of the fabricated beamforming $8 \times 10$ partial H-plane waveguide slot array antennas using an FSS. To verify the performance of the proposed beam-steering array antenna, we simulate and measure the reflection coefficient, far-field radiation pattern, peak gain, and aperture efficiency. The measured reflection coefficients at $−40°$, $−20°$, $0°$, $+20°$, and $+40°$ are $−9.1$ dB, $−8.2$ dB, $−10.3$ dB, $−7.7$ dB, and $−9.3$ dB at the resonance frequency, respectively. The designed FSS does not have 100% transmittance performance and is located close to the slot array antenna. Thus, the impedance matching of a total beamforming antenna system is not sufficiently good. It is expected to improve the reflection coefficient by an FSS with a perfect transmittance. The far-field radiation pattern and peak gain are measured in the full anechoic chamber system. The anechoic chamber is composed of a shield enclosure (size: 4 m × 2.5 m × 2.5 m), 18-inch pyramidal absorber, network analyzer, wireless communication test set, positioner, turn table, and dual-polarized transmit antenna. Figure 8 shows the full-wave simulated and measured far-field radiation pattern in the $x$–$z$ plane. The measured results are in good agreement with the simulated results. The measured peak gains at $−40°$, $−20°$, $0°$, $+20°$, and $+40°$ are 21.2 dBi, 23.6 dBi, 24.3 dBi, 23.6 dBi, and 21.3 dBi, respectively. The measured radiation patterns, including a forming angle, are well matched with the simulated results, even though the measured maximum peak gains are 1~1.5 dB lower than the simulated results. The discrepancy between both results is caused by the loss of a coaxial transition as a feeding structure. Moreover, the high aperture efficiencies at $−40°$, $−20°$, $0°$, $+20°$, and $+40°$ are measured as 44.2%, 62.7%, 69.2%, 62.2%, and 45.4%, respectively. Table 4 shows the comparison of the performances between the previously published TAs and the proposed antenna. The single-source TA cannot

achieve a low profile to obtain a relatively high aperture efficiency. However, the phase gradient FSS for the beamforming multi-source TA can be located close to multi-sources with a high taper and spillover efficiencies because of the uniform power distribution and the proximity of the FSS and sources, respectively. Therefore, using multi-sources rather than a single source is more appropriate to achieve an extremely low profile and a highly efficient beamforming TA.

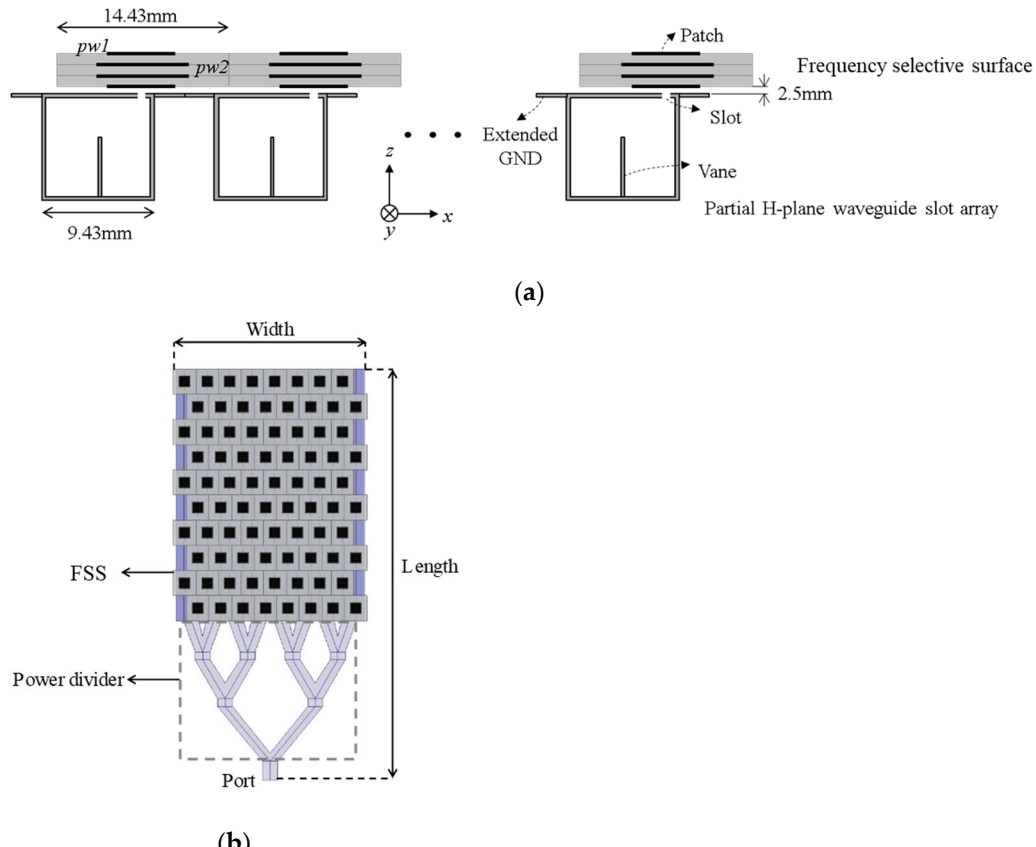

(**a**)

(**b**)

**Figure 6.** Structure of the beamforming partial H-plane waveguide slot array antenna. (**a**) Front view and (**b**) top view.

**Table 3.** Dimensions of the unit cell of the FSS for 8 rows against the beamforming angle (0°, +20°, and +40°) (Unit: mm).

| 0 degree | Row1 | Row2 | Row3 | Row4 | Row5 | Row6 | Row7 | Row8 |
|---|---|---|---|---|---|---|---|---|
| $pw_1$ | 6.3 | 6.3 | 6.3 | 6.3 | 6.3 | 6.3 | 6.3 | 6.3 |
| $pw_2$ | 7.1 | 7.1 | 7.1 | 7.1 | 7.1 | 7.1 | 7.1 | 7.1 |
| +20 degree | Row1 | Row2 | Row3 | Row4 | Row5 | Row6 | Row7 | Row8 |
| $pw_1$ | 0.1 | 5 | 6.9 | 6.9 | 6.9 | 3 | 6 | 6.6 |
| $pw_2$ | 0.1 | 5 | 6.9 | 7.3 | 7.5 | 3 | 6 | 7 |
| +40 degree | Row1 | Row2 | Row3 | Row4 | Row5 | Row6 | Row7 | Row8 |
| $pw_1$ | 7.7 | 3.0 | 6.9 | 6.9 | 4.7 | 6.9 | 6.9 | 6.4 |
| $pw_2$ | 7.7 | 3.0 | 6.9 | 7.4 | 4.7 | 7.3 | 7.5 | 6.4 |

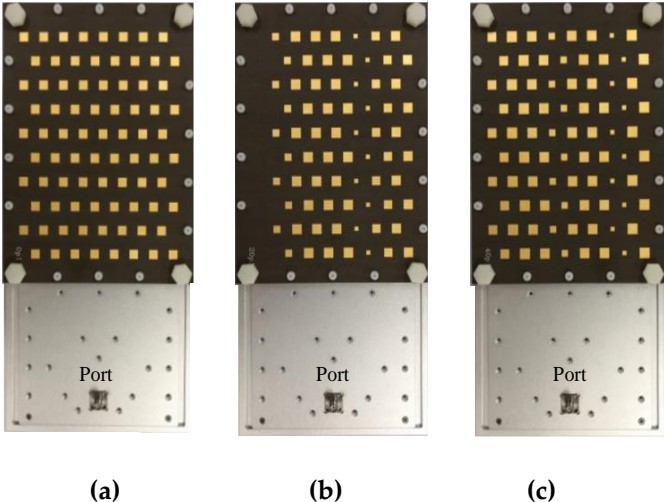

**(a)**                    **(b)**                    **(c)**

**Figure 7.** Photographs of the fabricated beamforming $8 \times 10$ partial H-plane waveguide slot array antennas using an FSS. (**a**) $0°$, (**b**) $+20°$, and (**c**) $+40°$.

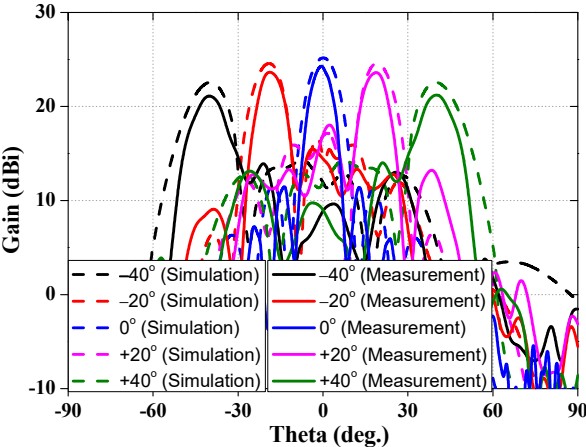

**Figure 8.** Full-wave simulated and measured far-field radiation patterns ($x$–$z$ plane).

**Table 4.** Comparison of the performances between the previously published TAs and the proposed antenna.

|  | Ref. [16] | Ref. [25] | Ref. [26] | Ref. [27] | Ref. [28] | This Work |
|---|---|---|---|---|---|---|
| Operation frequency (GHz) | 31 | 10.3 | 18 | 14.25 | 20 | 12 |
| F/D | 1.27 | 0.49 | 7.1 | 1.4 | 1.37 | 0.013 |
| Aperture efficiency (%) | 61.3 | 21.8 | 51.4 | 60.3 | 53.5 | 69.2 |

## 4. Conclusions

A 2D partial H-plane waveguide slot array antenna integrated with an FSS is proposed and designed to achieve a beamforming TA in this paper. The designed TA has a high taper efficiency by the multi-array sources and the compactness of the partial H-plane waveguide. Moreover, the proposed TA can inherently have a high spillover efficiency because the FSS cell is located just above the radiating slot, which can prevent the leakage of power. The FSS is designed by a square patch array based on a metallo-dielectric FSS and is formed by three dielectric layers with equally spaced capacitive printed patches. To verify its feasibility, three types of FSSs to operate a beamforming angle of $-40°$, $-20°$, $0°$, $+20°$, and $+40°$ are designed at 12 GHz. The F/D and aperture efficiency of the proposed antenna are 0.016 and 69.2%, respectively. From the measured results, the proposed TA is confirmed to have good beamforming characteristics and high aperture efficiency.

**Author Contributions:** Conceptualization, J.-G.L. and J.-H.L.; methodology, J.-G.L.; software, J.-G.L.; validation, J.-G.L.; formal analysis, J.-G.L.; investigation, J.-G.L.; writing—original draft preparation, J.-G.L.; writing—review and editing, J.-G.L. and J.-H.L.; visualization, J.-G.L.; supervision, J.-H.L. All authors have read and agreed to the published version of the manuscript.

**Funding:** This research was supported by the Basic Science Research Program through the National Research Foundation of Korea (NRF), funded by the Ministry of Education (No. 2015R1A6A1A03031833) and by the MSIT (Ministry of Science and ICT), Korea, under the Innovative Human Resource Development for Local Intellectualization support program (IITP-2023-RS-2022-00156361) supervised by the IITP (Institute for Information & Communications Technology Planning & Evaluation).

**Data Availability Statement:** Not applicable.

**Conflicts of Interest:** The authors declare no conflict of interest.

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
