# Peer review of "Low-Profile High-Efficiency Transmitarray Antenna for Beamforming Applications"

_electronics, doi:10.3390/electronics12143178_

Round 1

Reviewer 1 Report

The article is excellent in terms of novelty and significance. However, the introduction and the list of reference is inappropriate, because reflectarrays and wireless power transfer, are not in the scope of the article. I recommend removing references [1]-[17] and rewriting the introduction. Please focus on the waveguide slot antennas. Discuss each reference separately in the introduction. Among many papers on this topic, I recommend:  10.1109/LAWP.2017.2781262. Describe the works listed in Table 4.

A label is missing in Figure 3b. (gain?)

Please add a coaxial probe to Figure 2. 

Ccopyediting is required because some sentences don't read well, for example:

"In this paper, a low profile high efficiency TA for beamforming applications is pro-62 posed and investigated in this paper."

Correct typos, e.g. there should be a space between value and unit

In Figure 7 you shoud use the correct degree sign.

Reviewer 2 Report

1) In the paper beamforming applications is presented of transmit array antenna, In literature, various antenna with different method to reconfigurable beam and frequency are given. What is attractive about this research? Express the novelty of work with applications in abstract.

2) A Simple Monopole Antenna with a Switchable Beam for 5G Millimeter-Wave Communication Systems are given in literature with pin diodes for switchable beams. The design is simple and compact then your proposed work. How your work is attractive? And a review is performed about Metamaterials and Their Application in the Performance Enhancement of Reconfigurable Antennas. These research contain much information about antenna and beam forming.  

Provide in detail table of comparison of your work with literature work to show how your work is better and novel than other work. 

3) In section 3, the FSS is introduced. Introduce the is applications and basic working principle and formulation as Bandwidth and Gain Enhancement of a CPW Antenna Using Frequency Selective Surface for UWB Applications.

4) Rest of the paper is well presented. The above improvements are required. Also revise the paper to remove typo and improve English.

Thanks

Minor revision of English is required.

Reviewer 3 Report

The authors have described “Low profile high-efficiency transmit array antenna for beam- 2 forming applications”, my comments are as follows:

1.      What was the motivation behind choosing a partial H-plane waveguide slot array antenna?

2.      How has the phase of the incident wave changed?

3.      At what frequency, the H-plane waveguide is designed, and how it is optimized?

4.      Kindly add the surface current distribution of the H-plane and power divider.

5.      Kindly mention the design mechanism of the power divider.

6.      Kindly add the surface current distribution of the power divider.

7.      In Figure 3b, many side lobes are observed.

8.      Kindly add the gain in the Cartesian plot.

9.      In the literature, there are many simple beam-forming antennas observed, what was the major contribution of the authors?

10.   Which substrate is used? Why?

Round 2

Reviewer 2 Report

I am not satisfied with the authors comments. The introduction must be improved along with comparison. The table must contain the information of suggested antenna and antenna given in literature.

To prove novelty, show that the proposed results are best than litrature results mentioned in table.

Moreover, comments 1 & 3 of review round 1 is not addressed well

English must be improved, it is difficult to understand.

Round 3

Reviewer 2 Report

I am okay with comments now.